# SARS-CoV-2 Omicron Variant Genomic and Phylogenetic Analysis in Iraqi Kurdistan Region

**DOI:** 10.3390/genes14010173

**Published:** 2023-01-09

**Authors:** Sevan Omer Majed, Suhad Asad Mustafa, Paywast Jamal Jalal, Mohammed Hassan Fatah, Monika Miasko, Zanko Jawhar, Abdulkarim Yasin Karim

**Affiliations:** 1Biology Department, College of Education, Salahaddin University-Erbil, Erbil 44001, Kurdistan Region, Iraq; 2General Directorate of Scientific Research Center, Salahaddin University-Erbil, Erbil 44001, Kurdistan Region, Iraq; 3Biology Department, College of Science, University of Sulaimani, Sulaymaniyah 46001, Kurdistan Region, Iraq; 4Medical Lab., Technology Department, Kalar Technical College, Sulaimani Polytechnic University, Kalar 46021, Kurdistan Region, Iraq; 5Medical Analysis Department, Faculty of Applied Science, Tishk International University, Erbil 44001, Kurdistan Region, Iraq; 6Medical Laboratory Science, College of Health Sciences, Lebanese French University, Erbil 44001, Kurdistan Region, Iraq; 7Department of Biology, College of Science, Salahaddin University-Erbil, Erbil 44001, Kurdistan Region, Iraq; 8Department of Medical Microbiology, College of Science, Cihan University, Erbil 44001,Kurdistan Region, Iraq

**Keywords:** SARS-CoV-2, Illumina COVID-eq method, Omicron, Variants of Concern, receptor binding domain, GISAID

## Abstract

Omicron variants have been classified as Variants of Concern (VOC) by the World Health Organization (WHO) ever since they first emerged as a result of a significant mutation in this variant, which showed to have an impact on transmissibility and virulence of the virus, as evidenced by the ongoing modifications in the SARS-CoV-2 virus. As a global pandemic, the Omicron variant also spread among the Kurdish population. This study aimed to analyze different strains from different cities of the Kurdistan region of Iraq to show the risk of infection and the impact of the various mutations on immune responses and vaccination. A total of 175 nasopharyngeal/oropharyngeal specimens were collected at West Erbil Emergency Hospital and confirmed for SARS-CoV-2 infection by RT-PCR. The genomes of the samples were sequenced using the Illumina COVID-Seq Method. The genome analysis was established based on previously published data in the GISAID database and compared to previously detected mutations in the Omicron variants, and that they belong to the BA.1 lineage and include most variations determined in other studies related to transmissibility, high infectivity and immune escape. Most of the mutations were found in the RBD (receptor binding domain), the region related to the escape from humoral immunity. Remarkably, these point mutations (G339D, S371L, S373P, S375F, T547K, D614G, H655Y, N679K and N969K) were also determined in this study, which were unique, and their impact should be addressed more. Overall, the Omicron variants were more contagious than other variants. However, the mortality rate was low, and most infectious cases were asymptomatic. The next step should address the potential of Omicron variants to develop the next-generation COVID-19 vaccine.

## 1. Introduction

The development of the SARS-CoV-2 vaccine (severe acute respiratory syndrome coronavirus-2) is targeted at antibody production against the spike protein of the first SARS-CoV-2 initially discovered in 2019 that can induce immunity against the virus. Most vaccines that were developed early became a cornerstone for global public health and targeted the α variant [1]. However, the appearance of the Variant of Concern (VOC) continuously upregulated the virus transmission and escape of adaptive immunity [2]. The considerable variation of the SARS-CoV-2 variants increases the risk of the appearance of novel properties that may cause a threat to public health. The balance between transmission advantage and immune evasion has resulted in the evolution of five VOCs that display these characteristics to varying degrees. The α (B.1.1.7 in Pango nomenclature) and Delta (B.1.617.2) SARS-CoV-2 variants were the most transmissible variants worldwide. The α and Delta variants have a variation at position 681 for the furin cleavage site responsible for entry to the cells. Various studies showed that the α variants are more transmissible than the β variants but have less immune evasion than the Delta variant [2,3,4]. In contrast, both β (B.1.351) and γ (P.1) variants were known as strains that evade the immune system in vitro.

Omicron lineage (B. 1. 1. 529) is the fifth strain of VOC by the WHO. This strain was initially reported from South Africa on November 24, 2021 [5]. Since its discovery, this strain has been reported in more than 30 nations worldwide [6,7]. A new study in South Africa shows that Omicron has a three times more reinfection rate than Delta [7]. To understand and start explaining Omicron’s different or shared phenotype, sensitivity or resistance to existing vaccines, and whether Omicron-like strains that arise in the future may have heightened virulence, the first step is to identify its mutational profile carefully. Researchers found that about 50 mutations occurred in the Omicron genome, and 32 changes were found in the structure of the Omicron spike protein [8,9,10].

Omicron is divided into three sublineages BA.1, BA.2 and BA.3 [11]. The most transmissible sublineage is BA.1, whose spike glycoprotein encodes 30 amino acids related to the α variants; 15 are located in the RBD (receptor-binding domain) that can react with the ACE2 receptor, and the remaining are located in the RBM (receptor-binding motif) [12]. The high-affinity binding to ACE2 is doubled in Omicron compared to the Wuhan variant due to six mutations (G339D, N440K, S477N, T478K, Q498R and N501Y) in the RBD of the spike region [13,14]. Moreover, seven other mutations (K417N, G446S, E484A, Q493R, G496S, Q498R and N501Y) in the Omicron RBD are weakened binding to the antibody [15]. Furthermore, deletion (69–70, 143–145 and 211 amino acids) and insertion at site 214 are also detected in the NTD (amino-terminal domain) of the spike region of Omicron, which enhanced fusion and spike cleavage in the virion [16] and antibody neutralization [17], respectively. The “PRRA” spike protein insertion in the S1/S2 cleavage site offered the polybasic furin cleavage site that matches the RRARSVAS peptide in human ENAC-α of the most functionally significant changes in the evolutionary history of SARS-CoV-2 to date.

The objective of the current investigation was to identify the amino acid substitution and characteristics of the Omicron variation that may be connected to transmissibility and immune evasion. We examined the whole genome of the Omicron variant isolated from the Erbil, Kurdistan region of Iraq and compare it to other Omicron variants from other Iraqi cities previously published.

## 2. Materials and Methods

### 2.1. Ethics Statement and Consent to Participate

The study was approved by the Human Research Ethics Committee (HREC) at Salahuddin University-Erbil (Reference No. 4d/153) for the use of swab samples before study initiation. The study followed the ARRIVE guidelines 2.0, and all participants gave written informed consent and permission for publication. By signing the confirmed agreement of all research patients, permission for publication was acquired. All methods were carried out per the Helsinki Declaration of 1964, and all research patients gave written informed consent and authorization to be published.

### 2.2. Sample Collection

In this research study, 175 cases participated and were positive for the SARS-CoV-2 variants based on the RT-qPCR results. The nasopharyngeal/oropharyngeal specimens were collected from December 2021 to January 2022 at West Erbil Emergency Hospital. The collected samples were transported in a box with an ice seal to a public center laboratory (Hawler Teaching Hospital) and immediately used for viral nucleic acid extraction. Simultaneously, information on these Omicron cases’ clinical characteristics was collected by enquiring at West Erbil Emergency Hospital.

### 2.3. RNA Isolation and RT-PCR for Omicron Identification

This study used 200 μL viral samples from 175 samples to extract viral RNA, as described in the Nucleic Acid Extraction Kit (Magnetic Bead Method, REF: B200-32). An instrument of the Zybio EXM3000 Nucleic Acid Isolation System (Cat No.: ZBI-EXM3000, Thailand) was used to isolate and purify the viral RNA.

For the detection and identification of SARS-CoV-2 variants in these samples, the PowerChekTM SARS-CoV-2- S-gene Mutation Detection Kit Version 3.0 (Cat. No. R6922Q) was used. For preparing the RT-PCR mixture, the total reaction volume was 20 μL, consisting of 5 μL of template RNA, 10 μL of a 2X RT-PCR Master Mix and 5 μL of each Primer/Probe mix (FastPlex™ 1 Step SARS-CoV-2 Detection Kit/Catalog: 02.01.1020). The ZYbio program was according to the manufacturer’s manual for running real-time PCR. The reverse transcription step was set up to cDNA synthesis at 50°C for 15 min at one cycle, followed by initial denaturation at 95 °C for 5 min at one process. The thermocyclers were programmed for 40 cycles of denaturation at 95 °C for 10 ;s, followed by annealing at 60 °C for 30 s and elongation at 72 °C for 30 s.

### 2.4. Data Analysis

Fluorophore curves were evaluated on FAM, JOE, ROX, cy5 and Quasar. The instrument sets the threshold lines according to each Kit’s Ct value described on the CoA.

### 2.5. Whole-Genome Sequencing, Genome Assembly and Phylogenetic Analysis

In this study, sequencing processing was carried out for only three samples of 175 due to the high cost. Sample processing and whole-genomic sequencing were performed for the specimens with Ct < 30 [18].

The RNA molecules were extracted from three samples using the QIAGEN QIAmp Viral Mini Kit (PN 52904) and 250 ng of the extracted RNA from each sample was taken using the Ribo-Zero Gold rRNA depletion protocol to eliminate human ribosomal RNA (rRNA) (Illumina, 48 samples, Cat no. 20020598).

The RNA libraries were then constructed from reduced RNA by utilizing the TruSeq Stranded Total RNA kit (Illumina, Cat no. 20020599) and the IDT for Illumina TruSeq RNA UD Indexes (96 indexes, 96 wells) (Illumina, Cat no. 20022371). According to the TruSeq Stranded Total RNA, first- and second-strand cDNA molecules were synthesized using RT-qPCR-detected variant strains after the RNA molecules were fragmented. Adenylation was followed by adding poly (A) tail, adapter ligation was performed, and finally, amplification was completed [19]. Following amplification, the produced libraries were quantified, pooled and put into the Illumina NextSeq 500 method for sequencing. After that, they were normalized. The samples were sequenced using an Illumina NextSeq 500 system with a 150-cycle high-output kit (v2.5) to generate 75-bp paired-end reads. The results of the sequencing were saved for future research (Nextera). After the Bcl files were converted to fastq, the CLC Genomics Workbench version 11.0 was utilized to evaluate them (CLC, QIAGEN, Germany). Genetic mutations were confirmed and shown with the BAM files, applying Geneious software.

### 2.6. Bioinformatic Analysis

To perform read quality control, FAST-QC was used with the default parameters [20]. Eliminating adapter sequences and low-quality bases were conducted on the Fastp tool version 0.19 [21]. CD-HIT-DUP version 4.6.8 was used to filter out low-complexity reads and duplicates with fewer than 40 bases [22]. Off-target readings were then filtered out using Bowtie2 v2.3.4.3 and the SILVA database as a reference against the human genome version GRCh38.p13 [23].

The sequence readings were input for assembling viral genomes using the Wuhan-Hu-1 reference genome sequence (MN908947). Bowtie2 v2.3.4.3 was used to align the reads collected for each sample to the reference genome. SPAdes v3.14.0 was then used to perform de novo assembly using mapped reads. Consensus genome sequences were produced using the majority threshold criterion. In the studies, only sequence reads with a coverage level greater than 80% and a mean depth of 8 was employed.

On 7 February 2022, the whole-genomic sequence was submitted to GISAID (Global Initiative on Sharing All Influenza Data) to obtain the accession number. After the GISAID database staff confirmed the whole genome, the accession number (EPI_ISL_9606995) was provided. Now, the whole-genomic sequence is publicly available to all researchers in GISAID (https://www.epicov.org/epi3/frontend#32c498).

### 2.7. Phylogenetic Analysis

To find the closest relationship between our sample (virus name: hCoV-19/Iraq/Sevan/2022; Accession ID: EPI_ISL_9606995) and other SARS-CoV-2 variants in Iraq, the whole-genomic sequence of SARS-CoV-2 variants in Iraq was collected in GISAID (Global Initiative on Sharing All Influenza Data). The primary local alignment search (BLAST) tool in the GISIAD website searched for closely related sequences against all sequences in the EpiCoV database. About 500 genome sequences from the GISAID in different countries were retrieved and arranged to find the closest relationship between our genome and these sequences. Then, Molecular Evolutionary Genetics Analysis Version 11 (MEGA11) software was used to construct a phylogenetic tree.

## 3. Results

### 3.1. Clinical Data of Patients Infected with Omicron

Based on the RT-qPCR results, all the cases used in this study tested positive for SARS-CoV-2. Most patients were males below 50 years old. The clinical and demographic features are represented in Table 1. According to the questionnaire, the participants’ common symptoms were diarrhea, chills, fever and runny nose. Moreover, the infected patients were mainly not hospitalized, cured at home and not vaccinated.

### 3.2. The Mutational Change in the Omicron Variant in the Kurdistan Region of Iraq

Since its appearance in January 2021 and because of the low vaccination rate, the Omicron variant became dominant in 6 months in the Kurdistan region of Iraq. In order to determine the SARS-CoV-2 sublineages, whole-genome sequencing was carried out for three samples among 175 samples. The analysis of the genomic sequencing showed that they belong to the BA.1 sublineage (Figure 1). The genome analysis of the Omicron variant from the Kurdistan region shows one insertion in the spike region (214EPE) located in the RBD region, the binding site to ACE2. Six deletions were detected in the N-terminal region of the spike region (H69, V70, V143, V144, V145 and N211). Most substitutions were detected in the spike regions with 30 amino acid substitutions (A67V, T95A, G142D, L212I, G339D, S371L, S373P, S375F, K417N, N440K, G446S, S477N, T478K, E484A, Q493R, G496S, O498R, N501Y, Y505H, T547K, D614G, H655Y, N679K, P681H, N764K, D796Y, N856K, Q954H, N969K and L981F). One amino acid was substituted in the envelope region T9I, three amino acids in the membrane (D3G, Q19E and A63T) and three (P13L, R203K and G204R) were substituted in the nucleocapsid. Three amino acids were deleted in the nucleocapsid region (E31, R32 and S33). Many amino acids were deleted in the open reading frame (ORF) sequences with ten deletions (A86, S1265, L105, S106, G107, C209, I210, R233, R252 and Y502). Thirteen amino acids were also substituted in the ORF region (M87Y, L260P, F283S, P323L, I42V, K384R, L1266I, A1892T, T492I, P132H, I189V, Q208R and Y234H).

### 3.3. Quantitative Accumulation of Variations in the IKR Omicron Variant (B.1.1.529 + BA.1)

The amino acid changes in the Omicron variant spike protein may help reduce or prevent some of the deadliest symptoms, including pulmonary oedema, that leads to severe viral infection. The amino acid substitutions in the Omicron variant’s structure from Erbil city in Iraq were compared to the reference genome (hCoV-19/Wuhan/WIV04/2019) (see Table 2). The mutations have a phenotypic effect on the viral penetration, infectivity and transmissibility of the virus, and some have an indel effect. However, many variations of their impact have just been discovered. The 37 amino acid changes were also shown as colored balls in the spike sequence and structure (Figure 1b,c). These spike-related changes may damage cell barriers that line the inside of blood vessels within organs of the body, such as the lungs, leading to what is known as a vascular leak.

### 3.4. Spike Sequence Analysis of the Omicron Genome

Designing a structure based on the sequence of the Omicron variant showed that the most substitution was located in the spike region (Figure 2). The substitution D614G in the RBD is the standard, typical acid of all B.1 lineages.

### 3.5. Dominance of the Omicron Variant among the Kurdish Population during 2021

The phylogenetic analysis of the Omicron variants showed that infection by the new strains was raised significantly in most cities of the Kurdistan region during 2021. Since 2022, it has become the stand-alone variant for the disorder (Figure 3a). Moreover, the analysis showed that the variants differed from South African variants. The variants from Kurdistan of Iraq were more similar to the Omicron variants from Turkey (Figure 3b).

## 4. Discussion

The annual number of infected cases remained high despite international efforts to produce therapies and provide vaccines, partly due to the ongoing appearance of SARS-CoV-2 variations [5]. Although the Delta variation remains the most prominent VOC worldwide, the Omicron variants have proven to be the next dominant VOC [8]. The emergence of new variants of SARS-CoV-2 urged the need for comprehensive studies of the genomic structure of the virus globally [10]. The viral RNA is highly prone to variations and mutations when it steadily spreads among populations. Host-to-host travel and the stress experience of the virus in the host could lead to the risk of viral mutation and hence more aggressiveness and faster spread. With its specific context and social structure and the fluctuation of viral infection rates and mortality, the Kurdistan region of Iraq urgently needs to scan the infection rate and sequence the causative viral agents. Deep insight into the viral genome is a crucial necessity to track the phylogeny and origin of the virus variants that have spread in Kurdistan. This is in addition to the futuristic prediction of further possible changes in the viral genetic material. This is the first study to cover a randomized trial (the sample selected randomly for sequencing) of viral sequencing in three major cities with more than 200 samples to be sequenced via following-generation sequencing protocols. The outcome of the study can provide for the first time a robust data bank. This will offer insights to the stakeholders to further investigate the virus to predict future variations and strategies for fighting the viral spread. The structural alignment in this study showed several variations in the Omicron variants in the spike region, which may impact the ACE2 binding to the antibody. This may raise the question of the infection severity due to this variant and protection against the existing vaccines. These findings can arm the health authorities and anti-COVID approaches with endowed knowledge for preparedness and strategies for halting future viral outbreaks.

The genome analysis of the Kurdistan strains showed an increasing amino acid insertion, substitution and deletion in the spike region. These mutations may impact infectivity and transmissibility. Remarkably, the density of the transformation in the spike prion suppresses the function of this prion. On the one hand, the considerable variation in the spike region of the Omicron loss of the mRNA’s hairpin conformation leads to the genome’s fragility [24]. The Q493R mutation in the spike gene overlaps with many monoclonal binding sites [25]. Remarkably, some substitutions (N501Y, D614G, K417N and T478K) were concerned about reinfection and resistance to the presence of vaccines (Araf et al., 2022). These substitution mutations, G446S, Q493R, G496S, Q498R and Y505H, overlapped with the ACE2 antibody binding (Li et al., 2021). There are many substituted mutations detected in the spike (S1 and S2) gene (K417N, N440K, G446S, S477N, T478K, E484A, Q493R, G496S, O498R, N501Y, Y505H, P681H, N764K, D796Y, N856K, O954H, L981F, Insertion at 214–215) (Wang and Cheng, 2022) which were related to the immune escape and impact on binding to ACE2 (Li et al., 2020). Moreover, both K417N and T478K substitutions share Delta and Omicron variants, impacting immune tolerance and escape [15]. The previous study detected a deletion in the Omicron variant at the spike region G142. In contrast, in this study, there is a substitution from Glycine to Aspartic acid, and the impact of this amino acid should be addressed in detail.

Interestingly, there are nine other novel substitutional mutations (G339D, S371L, S373P, S375F, T547K, D614G, H655Y, N679K and N969K) which were detected in this study. Moreover, many deletions (A86, S1265, L105, S106, G107, C209, I210, R233, R252 and Y502) were also detected in this study, which was not mentioned before. In contrast, previously mentioned variations in the Omicron variants are not seen in the Kurdistan Omicron variants, which were ORF1a, (K856R, S2083del, L2084I, A2710T, T3255I, P3395H, L3674del, S3675del, G3676del and I3758V), ORF1b (P314L and I1566V), NSP9b (P10S, E27del, N28del and A29del), nucleocapsid (P13L) genes and spike (T95I and Y145D) [26,27].

Most of the studies revealed that the Omicron variants are less infectious than the other variants of SARS-CoV-2 but are more contagious [24]. The high transmissibility was shown in Figure 3 among the Kurdish population, which, since its appearance and nine months, the Omicron variants became the stand-alone variants despite the vaccination. As Acevedo et al. (2022) mentioned, different variants of SARS-CoV-2 have other impacts on eliciting and producing neutralizing antibodies. Based on this, the vaccines will impact each in another way. The monophyletic clade was shown during the genome analysis of Kurdish Omicron variants [28]; moreover, based on the previous report, the SARS-CoV-2 Omicron variants formed 20B clades and two subclades [26].

## 5. Conclusions

Omicron variants are dramatically more contagious than α variations, as seen by the enormous number of mutations in these variants. This demonstrates the fundamentals of epidemiologic science that could affect the end of the viral epidemic. According to the published data from the Ministry of Health of the Kurdistan Regional Government in Iraq, this may also force the virus to adapt to the necessity of the helper viruses to survive and decrease the rate of death by this type. The requirement for developing a new vaccine based on the novel mutations of the spike gene and the Omicron variants’ resistance to the existing vaccines must be determined by further research.

## Figures and Tables

**Figure 1 genes-14-00173-f001:**
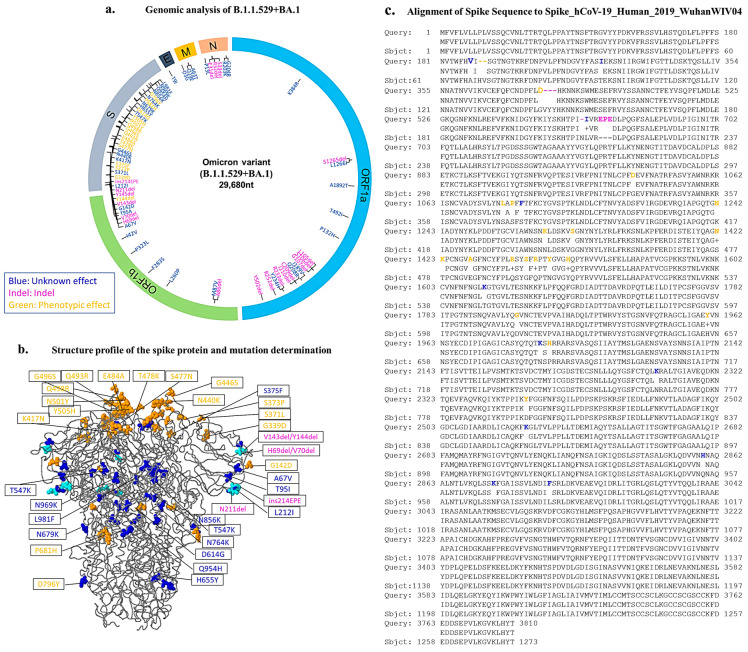
Full-length genome structure of the IKR Omicron variant (B.1.1.529 + BA.1). (**a**) The whole-genome map showed the regions (including ORF1a, ORF 1b and S = spike protein, E = envelope, M = membrane and N = nucleocapsid) and the location of 68 mutations, including deletion, insertion and substitutions. (**b**) The spike protein structure revealed 35 alterations in the subunits of S1, S1/S2 and S2. (**c**) Spike protein sequence which showed the location of amino acid changes, including deletion, insertion and substitutions was aligned to the spike protein sequence of Spike_hCoV-19_Human_2019_WuhanWIV04.

**Figure 2 genes-14-00173-f002:**
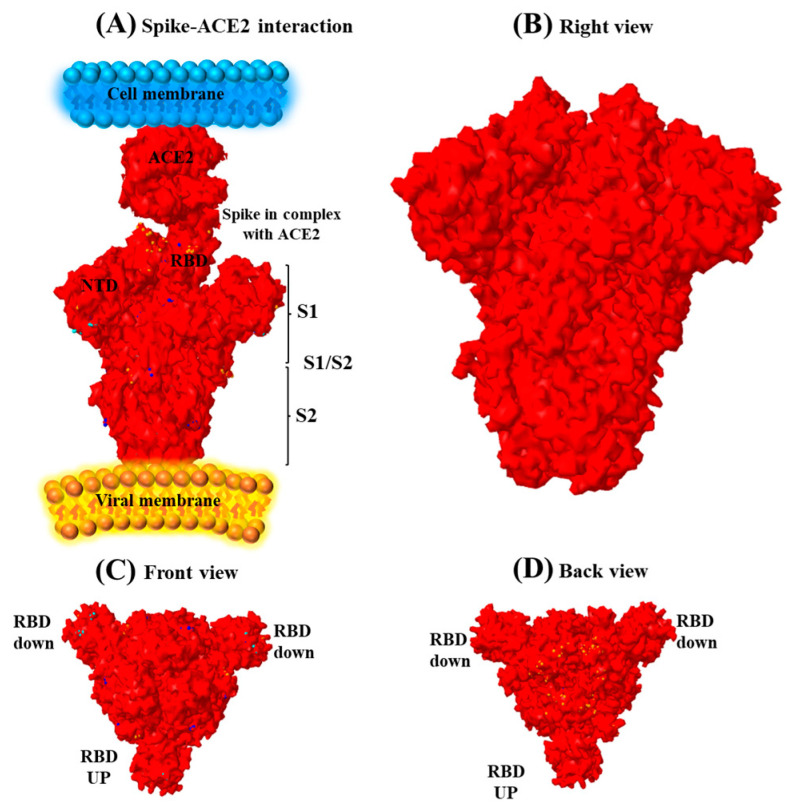
Structure of spike protein. (**A**) Different parts of spike protein include the N-terminal domain (NTD), receptor-binding domain (RBD), and S1 and S2 subdomains. The spike protein is attached to the ACE2 receptor by RBD. (**B**) Right view of the spike protein. (**C**) Front view of the spike protein. (**D**) Back view of the spike protein.

**Figure 3 genes-14-00173-f003:**
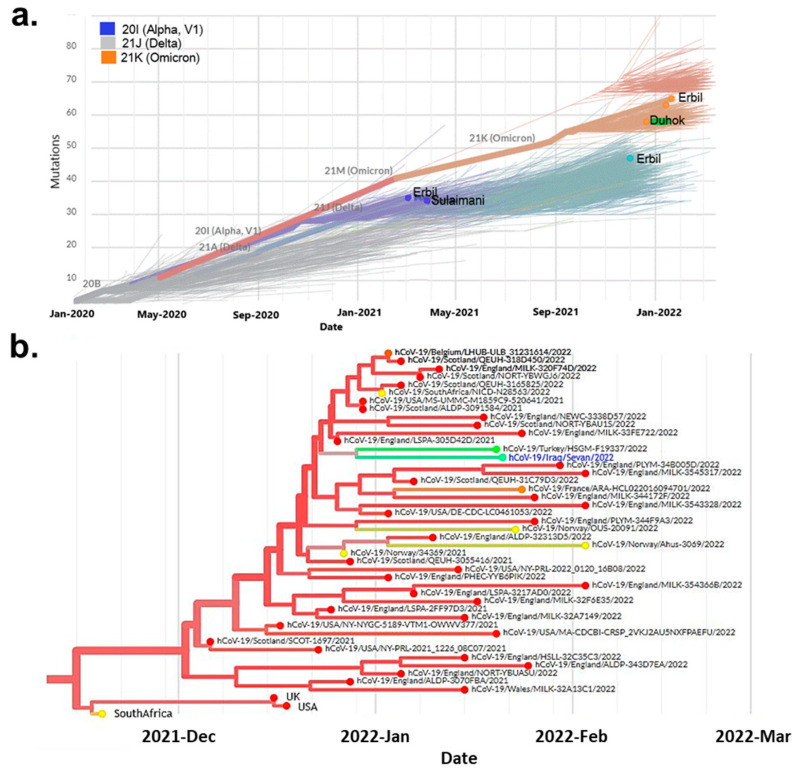
Maximum likelihood phylogenetic tree for the genomic sequences of SARS-CoV-2 variants recorded in Iraqi Kurdistan governates. (**a**) Analysis of eight genomic sequences recorded in the governates of Erbil, Sulaimani and Duhok. (**b**) Phylogenetic tree of the Omicron variants spreading among different countries. It contains 500 genomic sequences from December 2021 to March 2022 and is colored according to the nations. hCo-19/Iraq/Sevan/2022, colored in blue, is closely related to hCo-19/Turkey/HSGM-F19337/2022.

**Table 1 genes-14-00173-t001:** Clinical features of patients with the Omicron variant.

Clinical Features	No. of Cases (%)	OR (95% C1)	*p*-Value for Interaction
**Gender** Male Female	105 (60) 70 (40)	8.21 (6.10–10.02) 6.03 (5.09–8.34)	0.02
**Age** 50< 50≥	110 (63) 65 (37)	9.43 (7.06–12.3) 5.32 (4.21–6.89)	0.01
**BMI Categories:** Underweight ≤18.5 Normal weight = 18.5–24.9 Overweight = 25–29.9 Obesity = BMI of 30 or greater	15 (9) 95 (54) 50 (30) 10 (7)	4.20 (3.11–5.98) 8.89 (4.21–11.20) 5.32 (4.45–6.64) 4.01 (3.34–5.34)	0.05
**Common Symptoms** Fever Chills Cough Sore throat Runny nose Diarrhea	120 (68.6) 149 (85.1) 129 (73.7) 97 (55.4) 127 (72.7) 170 (87.5)	7.34 (5.11–9.21) 6.54 (5.67–8.43) 8.31 (6.98–10.22) 5.67 (6.34–7.84) 6.32 (5.32–7.89) 9.45 (8.34–11.45)	0.02 0.21 0.42 0.31 0.23
**Vaccination** Vaccinated Non-vaccinated	65 (37) 110 (63)	5.84 (4.32–6.45) 8.83 (6.38–10.89)	0.01
Hospitalized with Omicron Non-hospitalized Hospitalized	125 (71) 50 (29)	11.1 (9.43–12.34) 6.34 (4.50–7.69)	0.05

BMI: Body Mass Index.

**Table 2 genes-14-00173-t002:** Determination of variations in the structural protein of the Omicron variant.

*BA.1 Structure*	*List of Amino Acid Changes*
** *NSP3* ** ** *NSP4* ** ** *NSP5* ** ** *NSP6* ** ** *NSP8* ** ** *NSP12* ** ** *NSP14* **	K38R, S165del, L1266I, A1892T T492I P132H L105del, S106del, I189V, Q208R, C209del, I210del, R233del, Y234H, R252del, Y253del, L206P A86del, M87V F283S, P323LI47V
** *Spike glycoprotein* **	Spike A67V, Spike T95I, Spike G142D, Spike L212I, Spike S375F, Spike T547K, Spike H655K, Spike N679K, Spike N764K, Spike N856K, Spike H954Q, Spike N969K, Spike L981F Spike H96del, Spike V70deI, Spike Y145del, ins214EPE, Spike N211del, Spike V143del Spike Y144del, Spike G339D, Spike S373P, Spike S375F, Spike S477N, Spike T478K, Spike E484A, Spike Q493R, Spike G496S, Spike Q498R, Spike N501Y, Spike Y505H, Spike D614G, Spike P681H, Spike D796Y
** *Envelope* ** ** *Membrane* ** ** *Nuclear* **	T9I D3G, Q19E, A63T P13L , E31del, R32del, S33del, R203K, G204R

Orange= Phenotypic effect, Pink=Indel, Blue=Unknown effect.

## Data Availability

The whole-genomic sequence is publicly available to all researchers in GISAID (https://www.epicov.org/epi3/frontend#32c498) under the accession number (EPI_ISL_9606995).

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
