# Peer review of "SARS-CoV-2 Omicron Variant Genomic and Phylogenetic Analysis in Iraqi Kurdistan Region"

_genes, 2023, doi:10.3390/genes14010173_

Round 1

Reviewer 1 Report

  1. Abstract

    1. The VOC definition has to be changed in line 18 & 19

    2. Wrong terminology “flow of immunity”  used in line 22

    3. The mutations are not novel and are common in variants of omicron

  1. Introduction

    1. Line 40 vaccines do not neutralize spike protein but are targeted to generate antibodies that bind to it. (Line 40)

    2. Citations are not provided for the statements made. Line 61 to 65 

    3. Line 81 to 83 the GISAID description is out of context in that paragraph.

    4. The objectives are not clearly mentioned.

    5. Line 89 to 90 seems like a conclusion statement.

  1. Methodology

  1. The primer kits used to amplify the RNA and c-DNA generation for whole genome sequencing are not mentioned.

  2. Line 102 says 200 samples, but in table 1 Gender (105 + 70) adds up to 185, similary the other parameters.

  3. Line 127 what is the meaning of “models”

  4. was any pipeline used, like the arctic, etc., for the bioinformatics analysis?

  5. From the 200 samples why only 1 sample was sequenced? There is no explanation for that.

  6. Line 144 wrong terminology used. Are you explaining consensus creation?

  7. No mention of details of sequences retrieved for comparative study

  8. Please describe how the novel mutations were identified

Results

  1. When only one sample was sequenced and analysed what is the relevance of table 1 and why statistical analysis has been carried out

  2. Wrong mention of the year “2021” in line 172.

  3. Figure 1C should be shown with the alignments with the reference sequence

  4. A table can be prepared with all the mutations, their impact (if any), and reference (if reported)

  5. Line 221 “Designing a structure based on the sequence of the Omicron variant”. What is the methodology for this

  6. Significance of OR and p-value in table 1

  7. Line 176 to 188, what is the methodology used  

Discussion

  1. Line 288, The study was conducted in 2022 at that time omicron was already predominant

  2. Line 295, reference?

  3. Line 302, I understand only 1 sample was considered for sequencing, how is it a randomised trial?

  4. Line 329 to 332, From one sample this cannot be defined. Your sample has many problems in its sequence as highlighted in the GISAID of your sample.

  5. Line 339 to 342, again there is a discrepancy related to the number of sequences considered for the study

Conclusion

  1. Line 351 to 354, The sentence is not clear. 

  2. Line 353, how do you conclude that there is a decline in the rate of death by this variant?

  3. In the introduction, (line 85 86) you mentioned you would be determining biological and antigenic features, it has not been addressed in the results or in the discussion and the significance of the same.

Author Response

The manuscript has been revised according to the reviewer's comments and suggestions

Reviewer 2 Report

The article by  Sevan Omer Majed and co-authors is devoted to an extremely interesting topic, the study of the genetic variability of the SARS-CoV-2 virus in Iraqi Kurdistan. This topic is important because the dynamics of the genetic variability of the virus was studied using the example of one region. It is important that a whole genome analysis was carried out in the work. The expansion of the Omicron variant was shown and the main genetic changes in the virus genome were identified.

Majors :

In the work, little attention is paid to the analysis of the obtained results. Despite the enumeration of nucleotide substitutions, insertions and deletions, their contribution is completely incomprehensible. I would like to see a quantitative characterization of the results obtained. Perhaps in the form of a table, perhaps in the form of a diagram. It was also interesting to find out the binding of the quantitative accumulation of mutations to time.

Minors:

The article contains errors in the name of the virus, for example page 10 line 342 (SARS-CoV2) or page 10 line 346 (Sars-CoV2).

Author Response

All of the reviewer's Comments and Suggestions have been taken inconsideration and corrected.

Reviewer 3 Report

Authors have investigated different strains of Omicron variants from different cities of the Kurdistan region of Iraq 21 to show the risk of infection and the impact of the various mutations on the flow of immunity and vaccination. Authors have compiled useful information for the scientific community, but revision is required for more impact of publication.

My suggestions are as follows:

  1. Authors should include the inclusion and exclusion criteria of the SARS-CoV-2 Omicron infected patients.
  2. Authors should mention period of sample collections, the sample collection time is very important to understand Omicron-variants distribution in Iraq in particular periods.
  3. The Omicron variant has high environmental stability, high resistance against most therapeutic antibodies, and partial escape neutralisation by antibodies from convalescent patients or vaccinated individuals. Authors may include clinical with mutations in Omicron-variants I tabular form.
  4. The line no. 39-41 should be modified to "The development of SARS-CoV-2 (severe acute respiratory syndrome coronavirus-2) vaccine is targeted to neutralizing the spike protein of the first SARS-CoV-2 initially discovered in 2019 that can induce immunity against the virus.
  5. The line no. 106-107 should be modified to "Simultaneously, information on these Omicron cases' clinical characteristics was collected by enquiring at West Erbil Emergency Hospital “.
  6. Please check once again properly for any flaws throughout the text, standard abbreviations, and references. 

Author Response

The manuscript has been revised according to the  reviewer's comments and suggestions. 

Round 2

Reviewer 1 Report

Abstract:

Line 67: The number of samples is still 200 but in methods, it has been revised to 175.

Materials and Methods: 

Table 1 is not of any significance because the final sample was sequenced in only 1. It can be moved to Supplementary material. You can instead add a table of the individual sample with its demographic and clinical details.

Results:

Describe how many samples out of 175 tested positive for SARS CoV2 and Add a sentence to justify why only one sample was sequenced.

Figure 1c has not been updated with the alignment to the reference  sequence

Author Response

Response to reviewer 1 has been attached.

Reviewer 2 Report

The provided version of the article is better than it was before.

Author Response

The manuscript has been improved and all the reviewer questions has been answered and corrected.

Reviewer 3 Report

Manuscript may be accepted.

Author Response

English language and style have been checked. The methods are adequately updated and described.